# Learning Interclass Relations for Intravenous Contrast Phase Classification in CT

**Raouf Muhamedrahimov**[*]                RAOUF@ZEBRA-MED.COM
**Amir Bar**[*]                AMIR@ZEBRA-MED.COM
**Ayelet Akselrod-Ballin**                AYELET@ZEBRA-MED.COM

## Abstract

In classification, categories are typically treated as independent of one-another. In many problems, however, this neglects the natural relations that exist between categories, which are often dictated by an underlying biological or physical process. In this work, we propose novel formulations of the classification problem, aimed at reintroducing class relations into the training process. We demonstrate the benefit of these approaches for the classification of intravenous contrast enhancement phase in CT images, an important task in the medical imaging domain. First, we propose manual ways reintroduce knowledge about problem-specific interclass relations into the training process. Second, we propose a general approach to jointly learn categorical label representations that can implicitly encode natural interclass relations, alleviating the need for strong prior assumptions or knowledge. We show that these improvements are most significant for smaller training sets, typical in the medical imaging domain where access to large amounts of labelled data is often not trivial.

## 1. Introduction

In classification, categories are typically represented using a one-hot encoding and treated as equally different from one-another. However, this neglects the relations that could exist between categories, which are often dictated by an underlying biological or physical process, such as the progression of a pathology (Choi et al., 2018; Grassmann et al., 2018; Kuang et al., 2019; Peterfy et al., 2004). A natural approach is to reformulate the task as an Ordinal Regression problem and incorporate the ordinal nature of categories in the learning process, more closely reflecting the reality of our data (Armstrong and Sloan, 1989; Harrell Jr, 2015; Herbrich et al., 1999; Beckham and Pal, 2017a; Diaz and Marathe, 2019; Chu and Keerthi, 2007; Herbrich et al., 1999; Gu et al., 2014). Recently, Diaz and Marathe (2019) approached this problem by mapping ground truth labels into a soft distribution over categories. While shown to be beneficial, this approach relies on prior knowledge about the ordinal relations between categories. In this study, we propose an extension of this framework and demonstrate that underlying ordinal relations can be learnt from data while simultaneously being incorporated into ground truth label representations.

Our study focuses on the vital clinical task of intravenous (IV) contrast enhancement phase classification in Abdominal CT. IV contrast administration and the resulting enhancement patterns are often critical in the diagnostic process in CT. Multiphase CT scans are acquired in distinct physiologic vascular time points after initial IV administration, such as the *non-enhanced*, *arterial*, *venous* and *delayed* phases (Choi et al., 2018; Bae, 2010;

---

[*] Contributed equally

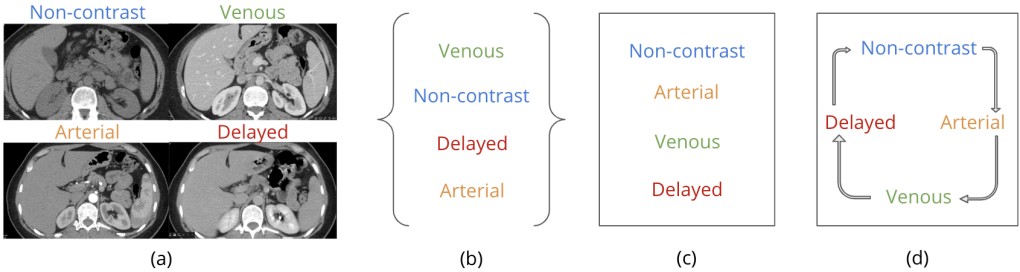

Figure 1: Examples of interclass relations for IV contrast phases. (a) An example of a CT abdomen image taken in four different phases. (b)-(d) illustrate different class relations where phases are (b) unordered, (c) ordered sequentially, and (d) ordered cyclically.

Guite et al., 2013). Scans acquired within a similar time-frames would reflect a gradual change in physiologic enhancement patterns and even share diagnostic features. Despite its clinical importance, information about the particular contrast phase of a CT series relies upon manual entry by a technician and is often partially or inconsistently captured in the DICOM metadata included with the scan. As such, an algorithmic solution to contrast phase classification is essential in permitting fully automated ML analysis of dynamic radiographic findings, capable of discerning, for example, between benign liver fibronodular hyperplasia and malignant hepatocellular carcinoma (Sun et al., 2017; Choi et al., 2018). Viewed as an ordinal regression task, phase relations are particularly interesting and can be understood either temporally (considering time-from-injection) or visually, where a cyclic pattern is observed – after the delayed phase the contrast agent is fully excreted.

Our main contributions are as follows. (1) We demonstrate that encoding cyclic ordinal assumptions into ground-truth label representations during training improves classification performance over a naive one-hot approach in a medical imaging setting of contrast enhancement phase classification. We show that these improvements are most significant for small training sets, which are typical in the domain. (2) We propose two reformulations of the classification task in which label representations are learned from data. Under constraints, we demonstrate that natural ordinal relations can be implicitly learnt and encoded during training, leading to the same improvements in performance while requiring few prior assumptions. (3) To the best of our knowledge, this is the first time a circular ordinal regression approach has been employed in the medical imaging domain of IV contrast.

## 2. Related work

**Ordinal regression** aims to predict the category of an input instance from a set of $K$ possible class labels in set $\mathcal{Y}$, while assuming a natural ordering (or ranking) between classes. This has been approached extensively from both as regression problem (Herbrich et al., 1999; Chu and Keerthi, 2007; Gu et al., 2014; Beckham and Pal, 2017b) and as a modified classification task (Cao et al., 2019; Frank and Hall, 2001). This study takes the latter approach and builds on recent work by Diaz and Marathe (2019), who propose the Soft Ordinal vectors (SORD) framework, where known ordinal information is encoded into the ground truth vectors through a soft target encoding scheme. Unlike previous work,

we apply this in the medical domain, and our goal is to encode ordinal assumptions based on visual semantics. We extend their framework and propose an approach we term PL-SORD, where we allow the network to implicitly choose an optimal label encoding through reformulation of the training loss, defining a set of "candidates" for the categorical ordering.

**Intravenous contrast** Recent works have modeled this as an image classification task, relying on human based annotations for training (Philbrick et al., 2018; Dercle et al., 2017; Ma et al., 2020). Dercle et al. (2017); Ma et al. (2020) propose systems to quality assess whether a scan was accurately taken in Portal Venous Phase (PVP). The proposed approach in Dercle et al. (2017) is semi-automatic and requires an expert in the loop. Philbrick et al. (2018) deals with contrast phase classification with a view of a full abdominal CT slice, however, primarily to evaluate neural networks visualization approaches in the context of clinical decision making. Instead, our primary focus is in demonstrating the effectiveness of introducing ordinal assumptions in label encodings. Our network is based on a 3D representation of abdominal organs using training labels extracted from scan metadata. In doing so, we are able to build up a training set of 264,198 full CT scans from 60 institutions, supporting the generalizability of our experiments.

## 3. Overview of approaches

While the experiments in this study focus on IV contrast, we present our approach more generally to highlight its broader applicability to any task where some relations exist between categories. Classification tasks performed over a set of discrete categories $\mathcal{Y}$, generally necessitate a label encoding $f : \mathcal{Y} \Rightarrow \mathbb{R}^{|\mathcal{Y}|}$, which maps any target class $t \in \mathcal{Y}$ (the ground-truth) to a vector of probability values $y_{\cdot|t} = f(t)$. The labels in this setting represent a probability distribution that the network attempts to match by optimizing some loss metric $\mathcal{L}$ over a given training set. Naturally, any relation that might exist between a class $i \in \mathcal{Y}$ and the target class $t$ can be represented in the encoded value $y_{i|t} = f_i(t)$, where $f_i(t)$ is the value for the category $i$ in the target vector $y_{\cdot|t} = f(t) \in \mathbb{R}^{|\mathcal{Y}|}$. Classification tasks over $K$ classes are most commonly represent each category as a one-hot vector (see Figure 2), where $y_{i|t}^{oh} = f_i(t) = 1$ if $i = t$ and 0 otherwise. In training, the network will treat all classes where $i \neq t$ as equally (or infinitely) wrong. In practice, some classes could be considered more correct than others. As such, there is motivation to define $f$ in a way that assigns a higher (non-zero) value to those categories closely related to a target class.

In Section 4, we start by formulating the task as an ordinal regression problem by assigning $K$ ordinal positions (or ranks) corresponding to each class, $\Lambda = \{r_1, r_2, ..., r_K\}$ in some $d$-dimensional continuous domain $r_i \in \mathbb{R}^d$. Using the SORD encoding approach, we incorporate known class relations into label representations based on a pairwise interclass similarity metric $\phi(r_t, r_i)$, which represents the ordinal "distance" between the categories. Within this framework, we can represent the cyclic nature of the categories by simply ranking them in polar coordinates. Once the categorical ranks $\Lambda$ and distance metric $\phi$ for the task are specified, the predetermined mapping $f$ can be directly applied to the labels.

In Section 5, instead of predefining a label encoding based on prior knowledge, we propose a more general approach whereby the label encoding $f$ is learnt from the data as part of the training process, exploiting ability of deep neural networks to generalize

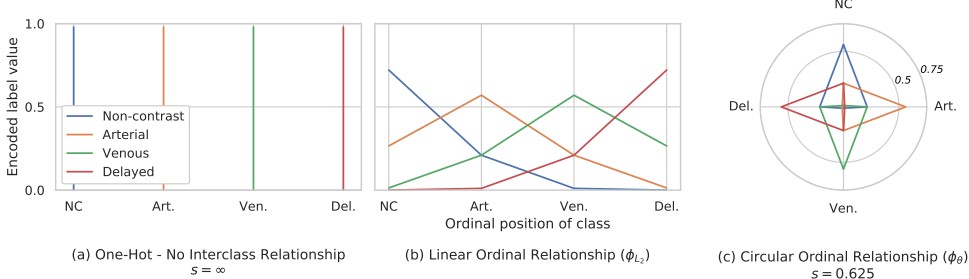

Figure 2: Examples of SORD-encoded label representation values under different assumptions on the relations between classes: (left) no relations (one-hot encoding, $s = \infty$), (middle) linear relations, and (right) circular relations ($s = 0.625$).

based on the visual semantics of the ordinal categories (Zhang et al., 2016; Maaten and Hinton, 2008; Chen et al., 2020; Zeiler and Fergus, 2014). In Section 5.1, we propose an approach we term PL-SORD, where we allow the network to implicitly choose an optimal label encoding $f$ from a set of "candidates" $\Lambda \in S$ for the categorical ordering. Extending this and loosening our constraints further, in Section 5.2, we propose a formulation of the problem that attempts to directly learn a parametrized encoding function $f$ from the data, optimized jointly with the network parameters. We explore how this impacts the optimal label representation learnt from the data and the extent to which it reflects natural interclass similarities

## 4. Label encoding with known interclass relations

### 4.1. Known categorical relations

We define $K$ ordinal positions (or ranks) corresponding to each class, $\Lambda = \{r_1, r_2, ..., r_k\}$ in some continuous domain $r_i \in \mathbb{R}^d$. If the target class is $t$, the label representation for the class $i$ can be computed as the softmax over the pairwise interclass similarities:

$$y_{i|t} = f_i(t) = \frac{e^{-\phi(r_t, r_i)}}{\sum_{k=1}^{K} e^{-\phi(r_t, r_k)}} \tag{1}$$

where $\phi(r_t, r_i)$ is a metric function representing the ordinal "distance" between the categories. Once the categorical ranks $\Lambda$ and distance metric $\phi$ for the task are specified, the mapping $f$ can be directly applied to the labels.

### 4.2. Known circular relations

The gradual return to some original state can be accounted for by defining the ordinal categories as angles in polar coordinates (See Figure 2). The distance metric can then be defined as the shortest angular distance between two classes in this space:

$$\phi_\theta(r_t, r_i) = [s \cdot \min(\|r_i - r_t \mod 2\pi\|_1, 2\pi - \|r_i - r_t \mod 2\pi\|_1)]^2 \tag{2}$$

Here, $s$ acts as a scaling factor for the target distribution, treated as a network hyperparameter – as $s \to \infty$, the distribution becomes one-hot, as $s \to 0$, all $y_{.|t}$ become equal.

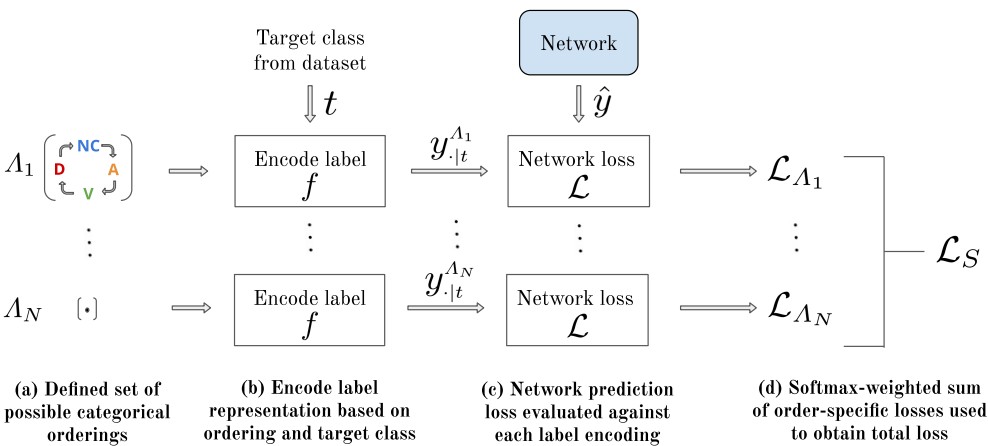

(a) Defined set of possible categorical orderings

(b) Encode label representation based on ordering and target class

(c) Network prediction loss evaluated against each label encoding

(d) Softmax-weighted sum of order-specific losses used to obtain total loss

Figure 3: Summary of the PL-SORD approach. A set of proposed categorical orderings define separate label encodings, each used to calculate an order-specific loss for a single network prediction. During training, both the network parameters and the weighted contribution of each loss term to the total are optimized over the data.

## 5. Learned label encoding

### 5.1. Learning the optimal categorical ordering

We relax our previous constraint and allow the network to choose from a set $S$ of possible categorical orderings (Figure 3). We refer to this approach as PL-SORD. While the set of possible relations $S$ can be defined quite generally, here we propose one possible approach. We first specify the set of unique ordinal positions that can be assigned to any of our $K$ categories, so any rank $r_i$ must have a value from $\{R_1, ..., R_M\}$. The set $S$ of possible orderings is then taken as all size-$K$ permutations of these "available" positions $\{R_i\}_{i=1}^M$. Using a fixed distance metric $\phi$ and Eq. 1, each ordinal permutation $\Lambda \in S$ defines a separate label encoding $y_{\cdot|t}^{\Lambda} = f(t; \Lambda)$ for each target class $t$, as in Section 4.1. This is then used to calculate the order-specific loss for the prediction $\hat{y}$, without changes to the network: $\mathcal{L}_\Lambda(\hat{y}, t) = \mathcal{L}(\hat{y}, y_{\cdot|t}^{\Lambda})$. The total network loss is computed as a weighted sum of the $N$ order-specific losses for the same network, as illustrated in Figure 3. So we optimize for both the loss weights $\lambda$ and the model parameters $W$ during training:

$$\min_{W, \lambda} \mathcal{L}_S(\hat{y}, y) = \sum_{j=1}^{N} \sigma_j(\lambda) \mathcal{L}_{\Lambda_j} \tag{3}$$

Here, $\sigma_j(\lambda) = \frac{e^{\lambda_j}}{\sum_{k=1}^{K} e^{\lambda_k}}$ is the $j$th output of the softmax applied over the learned weights $\lambda \in \mathbb{R}^N$, and represents the contribution to the total loss from permutation $\Lambda_j$. Intuitively, the ordinal permutation which contributes the lowest training loss would be assigned the largest weight during training, with the softmax ensuring this "choice" and a non-zero solution. We note that the training can also be performed in multiple steps, but observed no significant difference in the learnt loss weights $\lambda$ or final performance.

## 5.2. Directly learning label encoding

We now propose a formulation of the problem that attempts to directly learn the label encoding from the data. Specifically, when the target class is $t$, we hope to directly learn the distribution $y_{i|t}^{\alpha} = f(t, \alpha)$. It is clear that imposing no constraints would result in a degenerate solution, so we preserve the ground-truth signal by fixing the label value corresponding to the target class $t$ as $s \in (0, 1)$, which is treated as a hyperparameter, distributing the remaining $(1 - s)$ over the other classes. Concretely, the encoding for each target class $t$ is directly parametrized by $\alpha_t \in \mathbb{R}^{K-1}$ and given by:

$$y_{i|t}^{\alpha} = f_i(t; \alpha) = \begin{cases} s & \text{if } i = t \\ (1 - s)\frac{e^{\alpha_{t,i}}}{\sum_k e^{\alpha_{t,k}}} & \text{otherwise} \end{cases} \tag{4}$$

In training, we optimize both the network parameters $W$ and the encoding parameters $\alpha$:

$$\min_{W,\alpha} \mathcal{L}_S(\hat{y}, y) = \mathcal{L}(\hat{y}, f(y; \alpha)) \tag{5}$$

## 6. Experiments and results

### 6.1. Dataset

For this study, we use a proprietary abdominal CT dataset consisting of 334,079 scans (181,881 patients) from 2 healthcare systems and 60 institutions, including all cases that satisfy our study's inclusion criteria. All Patient Health Information (PHI) was removed from the data prior to acquisition, in compliance with HIPAA standards. See Appendix A for a detailed description of the dataset, sampling, and labelling. Approaches were compared on several randomly-sampled training sets with 2k, 4k, 8k, 16k, 32k, 80k, and 264k scans respectively. A validation set with 1,000 scans (963 patients) was used for model selection during training and fixed for all experiments. Both sets were assigned labels using rules-based regular expressions applied to the SeriesDescription DICOM tag. For final evaluation, 192 CT scans were manually labeled by an expert radiologist with similar representation of each class from 3 institutions. No patient was included in more than one set.

### 6.2. Model and training

The same model and training configuration was used for all experiments in this study. For each scan, a region of interest containing the liver, kidneys, aorta, and inferior vena cava (IVC), was automatically localized using an algorithmic approach similar to (Sahiner et al., 2018), ensuring the same anatomical region was extracted for all scans, independent of slice thickness and institutional protocol. 20 input slices were uniformly sampled from the extracted region and resized to $256 \times 256$. The input CT image pixel values were clipped to a CT window range centered at 40 with a width of 350 Hounsfield units. We use a 3D ResNet50 architecture (Hara et al., 2018; He et al., 2016) and apply a cross-entropy loss to the output of the model. Networks were trained on 2 NVIDIA GPUs using an Adam optimizer with a learning rate of $1 \times 10^{-4}$ and $\beta_1 = 0.9$, $\beta_2 = 0.999$ (Kingma and Ba, 2014). We use a batch size of 32 with equal representation of samples from each class. We continue each training run for 150k iterations, measuring performance on the validation set

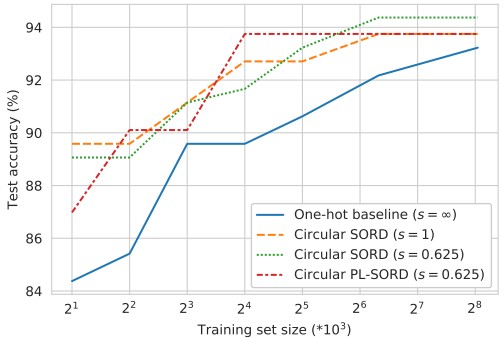

Figure 4: Comparison of contrast phase classification test performance using different methods. The highest test accuracy for a training set equal or smaller in size is reported. Ordinal formulations result in higher accuracy across all training set sizes, most notably for small datasets.

every 150 iterations, saving the model with the highest categorical accuracy. Models were implemented using Keras (Tensorflow backend).

### 6.3. Encoding known circular relationships

Introducing prior knowledge about class relations, we compare the performance of SORD encoding approach described in Section 4.2 to a one-hot baseline on the task of contrast phase classification. We start by assigning equally-spaced ranks $r_i \in (0, 2\pi)$ to each of the categories in order of their physiological appearance: $r_{NC} = 0$, $r_A = \frac{\pi}{2}$, $r_V = \pi$, and $r_D = \frac{3\pi}{2}$ and assume a circular relationship exists between phases, in line with their visual and diagnostic features. Two scaling factors for the hyperparameter $s$ defined in Eq. 2 are compared. Specifically, $s = 0.625$, for which the loss is more centered around the ground truth class and $s = 1$, where it is more distributed across adjacent classes. In Figure 4, SORD encoding resulted in performance improvements across all training set sizes for both scaling factors $s = 0.625$ and $s = 1$. We see the most marked improvements for small datasets. Specifically, when holding everything else fixed, the models utilizing the Circular SORD achieve improved performance over the standard one-hot formulation even when presented less than 10% of training data.

### 6.4. Learning the optimal ordinal encoding

We compare the approach described in Section 5.1 to a one-hot baseline and SORD encoding for contrast phase classification. A circular relationship is still assumed between phases and we allow them to take on 4 angular positions $r_i \in \{0, \frac{1}{2}\pi, \pi, \frac{3}{2}\pi\}$, without setting a constraint on the relative order. We then define our set of possible ordinal relations $S$ as all size-$K$ permutations of these equally-spaced positions. With an equal number of categories and possible positions, we are left with $|S| = \frac{(N-1)!}{2} = 3$ candidates for the natural ordering, after taking into account invariance to reversal and rotation in the circular setting[1]. Letting $\Lambda = (r_{NC}, r_A, r_V, r_D) \in S$ represent the assigned ranks of the non-enhanced, arterial, venous, and delayed phases, respectively, our set is given by:

$$S = \{\Lambda_1, \Lambda_2, \Lambda_3\} = \left\{(0, \tfrac{1}{2}\pi, \pi, \tfrac{3}{2}\pi), (0, \pi, \tfrac{1}{2}\pi, \tfrac{3}{2}\pi), (0, \tfrac{1}{2}\pi, \tfrac{3}{2}\pi, \pi)\right\} \tag{6}$$

---

1. In the angular setting, the permutations $(0, \tfrac{1}{2}\pi, \pi, \tfrac{3}{2}\pi)$, $(\tfrac{3}{2}\pi, 0, \tfrac{1}{2}\pi, \pi)$, $(\tfrac{3}{2}\pi, \pi, \tfrac{1}{2}\pi, 0)$ are all equivalent

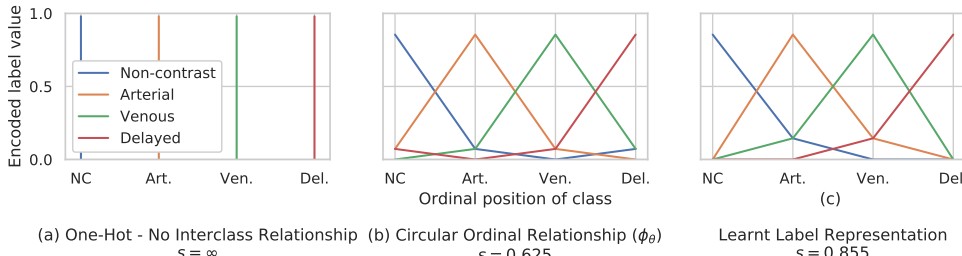

Figure 5: Comparison of data label representations with learnt encodings.

The results in Figure 4 indicate that the performance is comparable to the Circular SORD, although it requires fewer assumptions. By inspecting the learnt loss weights ($\sigma(\lambda)$ in Eq. 5), we found the permutation that minimizes the network training loss follows the natural ordering of the classes.

### 6.5. Directly learning label representations

 The approach described in Section 5.2 of directly learning label representations is compared to the one-hot baseline on the contrast enhancement phase classification task. We use the training set of 32k samples and set the hyper parameter $s = 0.855$ (see Section 5.2), such that we impose a maximum $y$ value which is similar to the circular SORD experiment. This model obtained a test accuracy of 92.28% whereas the one-hot baseline achieved 90.63%. We find the learnt label distribution $y_{i|t}^{\alpha}$ (in Figure 5c) results in an asymmetric encoding that could not be defined though the SORD framework described above, showing a higher degree of flexibility. It is clear that the easiest-to-optimize encoding does not necessarily reflect the natural ordinal relations without appropriate constraints on $f$, but rather weight is given to a single category responsible for the bulk of misclassifications for each target class. In additional to overlap of features, this highlights that some of the benefit of the SORD approach could be attributed to its accounting for biases, errors, or overlap in labels.

## 7. Conclusion

In this study we proposed three formulations of a classification task, centered around the introduction of interclass relations into label representations. For IV contrast phase classification in CT images, we demonstrate that incorporating cyclic ordinal assumptions during training significantly improves classification performance over standard one-hot approaches, particularly when datasets are small. In the PL-SORD approach, we show that we can learn a label encoding that implicitly incorporates the natural ordinal relations, leading to the same improvements in performance while requiring fewer prior assumptions. Finally, by alleviating the need to set any ordinal assumptions and directly learning a label encoding from data, we again demonstrate improvements over one-hot encoding. These approaches are very well suited for the medical domain, where small datasets and ordinal relations are common. Future work could explore the application of these approaches across more datasets and tasks, both within and outside the medical imaging domain.

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

## Appendix A. Dataset description

A summary of the dataset is included in Figure 6. For this study, we use a proprietary abdominal CT dataset consisting of 334,079 scans (181,881 patients) from 2 healthcare systems and 60 institutions, which is includes all cases that satisfy our study's inclusion criteria. Specifically, presence of the abdominal organs and an indication of phase in the DICOM SeriesDescription tag. The pool of patients was split into separate partitions – training with 264,198 scans (144,075 [79.1%] unique patients), validation with 35908 (19440 [10.7%] unique patients), and testing with 33,973 (18,366 [10.2%] unique patients). For training, all the cases in the training partition were used and further sub-sampling was performed to form the final validation and test sets. The validation set was used for metric evaluation during training and model selection. As such, the full partition was not used due to the inference time that would be required. Instead we performed a stratified sampling of cases, with 1,000 total scans (963 patients) with a balanced representation of each phase, equally sampled from 5 institutions. Both the training and validation sets were assigned labels using rules-based regular expressions applied to the SeriesDescription DICOM tag. For a held-out test set, 192 CT scans were sampled from the test partition and were manually labeled by an expert radiologist with similar representation of each class, with 3 institutions represented within each class. The goal was to ensure a consistent ground truthing approach that would not be subject to institutional bias or errors, as might be present in the DICOM-based labels. Modelling approaches were compared on several randomly-sampled training sets with 2k, 4k, 8k, 16k, 32k, 80k, and 264k samples respectively. All Patient Health Information (PHI) was removed from the data prior to acquisition, in compliance with HIPAA standards. The axial slices of all scans have an identical size of 512x512, but the number of slices in each scan vary between 42 and 1026, with slice spacing ranging from 0.45 mm to 5.0 mm. While it would be possible to assign the "leftover" patients to the training set after sub-sampling, we did consider the training set to be sufficient of size and thought it best to preserve the option would be available to perform further validation and testing with the remainder of those partitions, if needed.

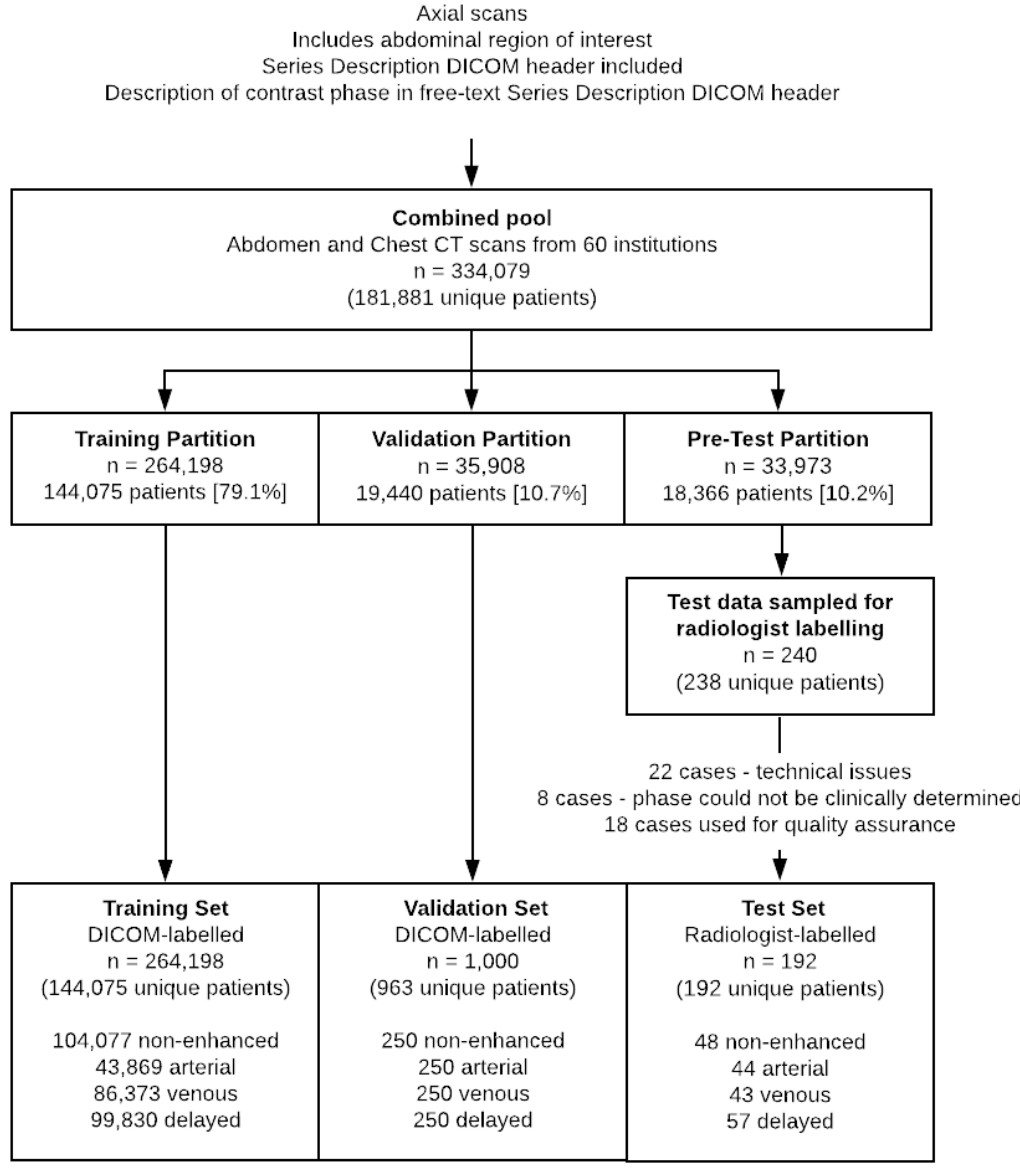

Figure 6: Summary of dataset and partitions.

## Appendix B. Performance of one-hot encoding, SORD, and PL-SORD

Testing accuracy for all encoding approaches and training set sizes are included in Table 1.

Table 1: Comparison of test accuracy for different ground-truth encoding approaches with varying number of training samples. All SORD-encoding results use a cyclic encoding.

| Training scans | Test Accuracy | | | |
|---|---|---|---|---|
| | One-hot $(s = \infty)$ | SORD $(s = 1)$ | SORD $(s = 0.625)$ | PL-SORD $(s = 0.625)$ |
| 2000 | 84.4% (162/192) | 89.6% (172/192) | 89.1% (171/192) | 87.0% (167/192) |
| 4000 | 85.4% (164/192) | 89.6% (172/192) | 89.1% (171/192) | 90.1% (173/192) |
| 8000 | 89.6% (172/192) | 91.1% (175/192) | 91.1% (175/192) | 90.1% (173/192) |
| 16000 | 89.6% (172/192) | 92.7% (178/192) | 91.7% (176/192) | 93.8% (180/192) |
| 32000 | 90.6% (174/192) | 92.7% (178/192) | 93.2% (179/192) | 93.8% (180/192) |
| 80000 | 92.2% (177/192) | 93.8% (180/192) | 94.4% (182/192) | 93.8% (180/192) |
| 264000 | 93.2% (179/192) | 93.8% (180/192) | 94.4% (182/192) | 93.8 % (180/192) |

