# OpenReview forum: "Learning Interclass Relations for Intravenous Contrast Phase Classification in CT"
_MIDL.io/2021/Conference — MIDL 2021_

### Official Review · AnonReviewer3 · 2021-03-01

**Confidence:** 5
**Preliminary Rating:** 1

**Summary:**

The paper presents a method to take advantage of domain knowledge of a task using the temporal nature of the CT acquisitions.

Claims:
"Encoding cyclic ordinal assumptions into ground-truth label representations during training improves classification performance"
"first time a circular ordinal regression approach has been employed in the medical imaging domain of IV contrast"


**Strengths:**

The paper addresses a common challenge faced when training models with labels which are coarse and could benefit from domain knowledge to more correctly define them.
...................................

**Weaknesses:**

The method seems overly complicated in it's presentation.
The evaluations could include more common baselines to justify the complication of this method.
The evaluations are not very convincing.
......

**Deanonymize Review:**

no

**Detailed Comments:**

Figure 1 didn't help me to understand the approach very well. I think it is important to make this approach very easy to understand if you would like people to use it.

The paper does not clearly support that this method is needed. I think more basic baselines should be evaluated. Such as using EMD: http://proceedings.mlr.press/v70/beckham17a.html or less formally just simply not penalizing neighboring predictions.

Why were linear ordinal relationships not used as a baseline in the experiments?

The evaluation merges data from multiple sources into one dataset and then evaluates without any sort of multiple trials. You could split the 10 sites into a 10 fold cross validation which would give you a more reliable estimate of if this method really works.

If your claim is that this works better, there should be some sort of significance test.

The claims of the paper involve this being applied to IV contrast but the focus of the discussion is not in the clinical importance. There should be more justification for how this added performance is necessary to the clinical tasks. Why does this work improve the clinical task?

One aspect that makes me confused about the value of this paper is that the authors have access to 100k+ CT scans but work on a method that benefits "small training sets, which are typical in the domain". The claims of the paper focus on the specific task of CT with IV contrast yet the title is general and says nothing about CT. So I find the story inconsistent.


**Justification Of The Preliminary Rating:**

Story not clear.
Claims not supported by experiments.

.................................................................................................................................................

**Paper Type:**

both

**Questions To Address In The Rebuttal:**

Clarify the story and contribution (make the title say what you do)
More baselines to justify this method.
Significance testing in experiments.
Justify clinical significance.


**Special Issue:**

no

---

### Official Review · AnonReviewer1 · 2021-03-07

**Confidence:** 4
**Preliminary Rating:** 3
**Recommendation:** Poster
**Final Rating:** 3

**Summary:**

This paper presents an experimental evaluation of concepts for embedding relationships between classes in a classification problem into the loss function. In particular, the paper proposes class weights based on distances in a polar coordinate system, which makes it possible to capture a circular dependency between classes. The presented application is classification of contrast-enhanced abdominal CT scans according to the contrast phase, which starts without any contrast enhancement followed by the enhancement increasing and eventually decreasing again, hence the circular order.

**Strengths:**

* There is substantial interest in encoding ordinal relationships between classes into the training process of neural networks, e.g., when training networks that classify images according to some kind of severity grade, which is essentially a hybrid between a classification task and a regression task. This paper investigates an elegant approach for circular dependencies.
* The experiments demonstrate that encoding prior knowledge about the order of the classes improves the performance, especially when a limited amount of training data is available.
* Next to explicitly encoding a specific circular relationship of the classes, the paper also investigates variants of the approach where the relative weights of the classes are learned as part of the training process.
* The approach is validated with almost 200 CT scans.

**Weaknesses:**

* The experiments with learning the optimal encoding are somewhat limited, only a single experiment was performed. Results for different training set sizes and different values for the parameter s would have been valuable.

**Deanonymize Review:**

no

**Detailed Comments:**

* The distribution of the dataset into training, validation and test partitions is stated as "90%/10%/10%" which would be 110% in total, this seems to be a typo.
* It would be useful to have the results presented in Figure 4 also in a table.

**Final Rating Justification:**

The paper presents an interesting idea. Even though some of the other reviewers are less convinced by this idea, I believe it would make a good contribution and would lead to useful discussions at the conference.

**Justification Of The Preliminary Rating:**

This paper presents a nice idea that is thoroughly evaluated on a large dataset. Especially the more flexible learning-based approaches might have several applications other than the presented application of contrast-phase classification.

**Paper Type:**

both

**Questions To Address In The Rebuttal:**

* Why is the test set so much smaller than the validation set (192 vs 1000 scans)?
* It seems a bit counterintuitive to train both the class weights and the classification network itself together. Can you imaging that it would make sense to approach this problem in two stages, where first the class weights are learned and then the classification network is trained?

**Special Issue:**

no

---

### Official Review · ~Jannis_Hagenah1 · 2021-03-07

**Confidence:** 4
**Preliminary Rating:** 1
**Recommendation:** Poster
**Final Rating:** 3

**Summary:**

In this paper, the authors apply the concept of ordinal regression to phase classification in abdominal IV CT. Thus, they collect a dataset and transfer the SORD framework to the given problem. Furthermore, they present a novel approach to learn the interclass relationship together with the network training in a completely data-driven way, extending the SORD framework. They show that both approaches outperform a classical one-hot-encoding approach on the given dataset, specifically when only a small portion of training data is used.

**Strengths:**

In general, the idea of incorporating interclass relationships is quite novel in the medical domain and might be very interesting in some areas as an alternative to standard classification. Additionally, designing a framework to learn the SORD relationships directly from data is challenging and I am impressed that the authors could make the network converge with such a large optimization within the loss function.
The collection of such a large dataset is definitely worth mentioning. Following the open science spirit of MIDL, I would highly recommend making the data publicly available and to include a link in the document.


**Weaknesses:**

Unfortunately, I have several concerns about the manuscript, and the main one brings me to the very basic assumption of the presented work:
Regarding IV CT phase classification, it is not clear to me why some phases are more similar to each other than others are. If the contrast bolus is in the arteries, they have a high contrast, and the veins have a high contrast when the bolus arrives there. I do not see why misclassifications between these images should be counted less harsh and why this should help the network to gain performance. Or, to reframe it: I doubt that there is some causal relationship between the different phase images that guides the network towards better generalization! Or is the bolus spreader that much over the vascular system that there are some "inaccurate states"? And why could that not be addressed via data curation?
To be fair, your results show the superior performance of the SORD approach. But this brings me to my second concern that is closely related to the first one: I am missing a discussion of the results! As stated above, I can imagine that the performance increase might not follow the class relationships but may be induced by adding additional gradient information to each class. As each datapoint now spans multiple classes, the SORD approach works like some form of oversampling or data augmentation, and hence provides higher robustness on small datasets. I am sure you have a great argument against this hypothesis, but this is exactly what I am missing in the discussion. Especially for such unintuitive results, an adequate discussion is a must!

Furthermore, I have several major concerns:
-The aim of the presented paper is a bit ambiguous and partially unclear. The title claims a very general methodological development, which should incorporate in-depth analysis of the performance and hyperparameter choices on several standard datasets. However, all results are computed on a dataset called “proprietary”, leaving the impression that the goal of the study is to apply a known computer vision method to IV CT phase classification. Please clarify the storyline!

-In 5.1, you state that “the ordinal permutation which contributes the lowest training loss would be assigned the largest weight during training”. I do not see why the model should increase any of the lambdas as the loss function is basically a weighted sum with the lambdas as weights. In fact, the best strategy for the model to decrease the loss is to set all lambdas to zero. How do you make sure that this will never happen? Did you assess the lambdas after training? Please comment on that!

-Unfortunately, the structure of the dataset remains unclear. The claimed 264,198 samples in the training set do not match 90% of the full dataset, consisting of 334,079 samples. It is not clear why only subsamples of the validation and test data were used. Additionally, it is not described whether the splits cared for patient distinction and/or center distinction through the sets. Hence, it is unclear whether data of the same patient could be in the training and the validation set, for example. Furthermore, I do not understand why the ground truth labels in the training dataset could be assessed by simply reading the DICOM header while the test data was “manually labeled by an expert radiologist”. If the header information is available, I would assume that this data is optimal and hence the expert is not necessary and, in the worst case, introduces human errors to the ground truth.
Please revise section 6.1 and make the data used in this study clear!

-Even though the paper is well written in general, the mathematical notation is partially insufficient and some variables appear without explaining what they are. Details on this can be found in the “Detailed Comments” section.


**Deanonymize Review:**

yes

**Detailed Comments:**

Besides the major points described above, I have some minor concerns:

-In section 3, you use $f_{i}(t)$ as a specific index of the resulting vector $y_{.|t}$. Please introduce this as in general, $f$ maps to $\mathbb{R}^{|\mathcal{y}|}$

-In the caption of Fig. 2, you are stating that $s=1.25\pi$ regarding the circular relations, while I assume that it is $s=0.625$.

-In 4.1, you use the dimensionality $d$ without denoting it.

-In 4.2, Equ. 2 is really hard to read. First, please use the standard multiplication sign to not confuse it with other operators. Additionally, the spacing before the mod operators are highly confusing. Please make this more concise.

-In 4.2, I do not get why the label distribution becomes one-hot if s is close to infinity as s is the same factor for all class pairs. With an infinite scaling factor s, $\Phi$ should become somewhat independent of the distance between $r_t$ and $r_i$, and hence I would expect the same value for each class. Could you please comment on that?

-In 5.1, you use $y^{\Lambda}$ without introducing it.

-In 5.2, reordering the sentence between equations 4 and 5 might help to understand it easier, e.g. “… both the network parameters W and the encoding parameters alpha”

-In 6.3, you reference to Fig. 6.3, which is non-existent. I believe this should be referencing to Fig. 4? Same holds for section 6.4

-In Fig. 4, there is a backslash in the y-label that should not be there. Additionally, the x-label unit is a bit strange to understand. Maybe replace it by something like $*10^{3}$.

-In 6.4, you state that the learned weights were assessed in a “manual analysis”. Please clarify what you mean by this.

-The paper is slightly too long. Please shorten it to the required 8 pages.


**Final Rating Justification:**

The author's rebuttal letter helped me a lot understanding the ideas behind crucial parts of this work and the proposed methods. I appreciate the new title and storyline, the paper's focus is now more clear.

All in all, the authors could convince me that the presented study is of interest for the MIDL community and could definitely complement the program. Hence, I am willing to adjust my rating to a weak accept and would be happy to see their work presented as a poster at MIDL 2021. However, I strongly recommend the authors to revise the manuscript according to the comments of all reviewers before final submission.

**Justification Of The Preliminary Rating:**

Even though the paper presents an interesting concept and contains neat ideas for data-driven parameter learning, I cannot vote for an accept in its current form. The lack of an adequate discussion reduces the scientific value significantly, specifically for results that are highly surprising and not intuitively expectable, at least from my perspective. Additionally, the focus of the paper is ambiguous and important details of the dataset remain unclear.
However, I would highly recommend the authors to keep up their research in this field as it might be quite interesting for other applications, for example optimal prosthesis type classification where ordinal relationships are intuitively present. Thus, it might be a good choice to submit a revised version to a journal that allows more pages to include in-depth evaluations of the methods performance as well as hyperparameter choices on multiple datasets, including typical benchmarks like the Adience Face Dataset (https://talhassner.github.io/home/projects/Adience/Adience-data.html) or, from the medical domain, the diabetic retinopathy dataset (https://www.kaggle.com/c/diabetic-retinopathy-detection/overview ). This would increase the scientific value significantly!


**Paper Type:**

both

**Questions To Address In The Rebuttal:**

Please comment on my main concerns as described in the "Weaknesses" section. Additionally, I have listed several points that would would improve the manuscript in the "Detailed Comments" section. Please also address them.

**Special Issue:**

no

---

### Official Review · AnonReviewer2 · 2021-03-08

**Confidence:** 4
**Preliminary Rating:** 3
**Final Rating:** 3

**Summary:**

The authors propose a method to utilize relationships between image classes to improve the classification accuracy.  Specifically they model the relationship between CT images taken at different phases of intravenous contrast flow.  They experiment with different ways to model these relationships using prior knowledge, or alternatively allowing the network to learn the relationship during training.  Their experiments determine improved classification accuracy when such relationships are modelled, especially when using smaller numbers of samples in training.

**Strengths:**

The paper is well written and clearly explained.  The previous literature is adequately discussed and the experimental results are clear.  I think that the subject of interclass relationships is a relevant and important one, since there are many conditions with very subtle differences in their appearance on imaging and few authors make use of this information.

**Weaknesses:**

The problem being solved does not seem to be particularly common or clinically relevant (see detailed comments below) and the authors do not address this or speculate about how the method could be generalized to more common/relevant tasks.

The dataset description is a bit unclear (see detailed comments).  It is always better to include a data table for clarity.


**Deanonymize Review:**

no

**Detailed Comments:**

The specific problem being solved by the authors does not seem to be particularly clinically relevant.  Although I am not familar with these contrast phased images, it seems that they can be easily distinguished using dicom headers and it is therefore only in less common scenarios (where dicom information is missing/incorrect) that such a method would be needed.

This paper also tackles only classes which have an ordinal relationship.  It seems to me that this is not a particularly common situation, although I am interested to hear the author's view on that.  I would be interested how the method could be extended to more general class relationships (not ordinal ones) such as classifying abnormal patterns with very similar appearances.

The dataset description is a bit unclear to me.  I would prefer to see a table.   Why does it say 90/10/10 split between train/val test (this does not add to 100!) .  I would rather see exact numbers in each set (and percentages in brackets after).   The percentages for val/test do not seem to correspond to the numbers mentioned subsequently.   It is not clear to me how all labels were assigned (the text seems to suggest that only a subset are labelled?)

There is a wide range of slice spacings (0.45mm to 5mm).  I am curious how this affects the learning process since the authors do not seem to compensate for this with any resampling etc.   Could resampling improve the process?

There are several references to Figure 6.3 , which I believe should say Figure 5.

**Final Rating Justification:**

I thank the authors for their response which has clarified a few issues, however it has not been sufficiently robust to improve my rating for this  paper.

**Justification Of The Preliminary Rating:**

I think this is an interesting method to solve the problem of labels with ordinal relationships.  The paper is well written and experiments are sound.   My doubts relate to the clinical relevance of the task tackled here, and whether it could be easily expanded to more common/relevant clinical tasks where label relationships could be utilized.

**Paper Type:**

methodological development

**Questions To Address In The Rebuttal:**

Please see "detailed comments" for questions I would like to be addressed.

**Special Issue:**

no

---

### Meta-Review · Area_Chair1 · 2021-03-28

**Recommendation:** Accept (Poster)

**Metareview:**

This is a borderline paper -- while the underlying idea is good and relevcant, the authors don't do a very good job of selling it; their experiments are performed on a very specific task with limited clinical relevance. The reviewers had a number of questions regarding experimental setup, which were largely answered in the rebuttal.

**Paper Type:**

both

---

### Decision · Program_Chairs · 2021-03-31

Accept